# sCD14-ST and Related Osteoimmunological Biomarkers: A New Diagnostic Approach to Osteomyelitis

**DOI:** 10.3390/diagnostics14151588

**Published:** 2024-07-23

**Authors:** Emanuela Galliera, Luca Massaccesi, Virginia Suardi, Elena de Vecchi, Francesca Villa, Zhang Yi, Guorui Suo, Arianna B. Lovati, Nicola Logoluso, Massimiliano M. Corsi Romanelli, Antonio V. Pellegrini

**Affiliations:** 1Department of Biomedical Sciences for Health, Università degli Studi di Milano, 20122 Milan, Italy; luca.massaccesi@unimi.it (L.M.); mmcorsi@unimi.it (M.M.C.R.); 2Cell and Tissue Engineering Laboratory, IRCCS Istituto Ortopedico Galeazzi, 20157 Milan, Italy; arianna.lovati@grupposandonato.it; 3 Laboratorio Sperimentale Ricerche Biomarcatori Danno d’Organo, IRCCS Istituto Auxologico Italiano, 20149 Milan, Italy; 4Chirurgia Ricostruttiva e delle Infezioni Osteoarticolari (C.R.I.O.), IRCCS Istituto Ortopedico Galeazzi, 20157 Milan, Italyantonio.pellegrini@grupposandonato.it (A.V.P.); 5Laboratorio di Analisi Chimico Cliniche e Microbiologiche, IRCCS Istituto Ortopedico Galeazzi, 20157 Milan, Italy; 6 Immunoassay Reagent Rand Department, Shenzhen Mindray Bio-Medical Electronics Co., Ltd., Shenzhen 211111, Chinasuoguorui@mindray.com (G.S.); 7Department of Experimental and Clinical Pathology, IRCCS Istituto Auxologico Italiano, 20149 Milan, Italy

**Keywords:** osteomyelitis, bone infection makers, diagnostic challenges, CD14-ST, SuPAR, circulating osteoimmunological markers, laboratory diagnostic technique

## Abstract

Osteomyelitis (OM) is a major challenge in orthopedic surgery. The diagnosis of OM is based on imaging and laboratory tests, but it still presents some limitations. Therefore, a deeper comprehension of the pathogenetic mechanisms could enhance diagnostic and treatment approaches. OM pathogenesis is based on an inflammatory response to pathogen infection, leading to bone loss. The present study aims to investigate the potential diagnostic role of a panel of osteoimmunological serum biomarkers in the clinical approach to OM. The focus is on the emerging infection biomarker sCD14-ST, along with osteoimmunological and inflammatory serum biomarkers, to define a comprehensive biomarker panel for a multifaced approach to OM. The results, to our knowledge, demonstrate for the first time the diagnostic and early prognostic role of sCD14-ST in OM patients, suggesting that this biomarker could address the limitations of current laboratory tests, such as traditional inflammatory markers, in diagnosing OM. In addition, the study highlights a relevant diagnostic role of SuPAR, the chemokine CCL2, the anti-inflammatory cytokine IL-10, the Wnt inhibitors DKK-1 and Sclerostin, and the RANKL/OPG ratio. Moreover, CCL2 and SuPAR also exhibited early prognostic value.

## 1. Introduction

Osteomyelitis (OM) is a major challenge in orthopedic surgery, characterized by inflammation of the bone and surrounding tissues due to bacterial infections, leading to bone destruction [1,2].

One of the primary pathogens responsible is *Staphylococcus aureus*, which can infect bone through the bloodstream, open fractures, or surgical procedures [3]. The presence of biofilm-forming bacteria on bone surfaces and implants complicates treatment due to their resistance to antibiotics and immune clearance. Diagnosing OM remains complex due to the limitations of various diagnostic tools. Imaging techniques such as X-ray and MRI often fail to provide a definitive diagnosis, especially in the presence of metal implants. Although bone culture is the gold standard, it has a false-negative rate of up to 40%.

A comprehensive diagnostic approach combining imaging and laboratory tests is essential [4]. However, routine laboratory makers of inflammation lack the sensitivity and specificity required for OM diagnosis [5]. Therefore, there is a critical need for specific circulating biomarkers that can offer more information.

The immune response to OM involves a complex interplay between pro-inflammatory and anti-inflammatory signals. The infection triggers an inflammatory response characterized by the release of cytokines such as IL-6 and TNF-alpha, which recruit immune cells to the infection site. OM is defined as an inflammatory condition of the bone, commonly caused by infection. The infection triggers an inflammatory response that disrupts bone remodeling, resulting in excessive bone resorption and subsequent bone loss. The interaction between the skeletal and immune systems, known as osteoimmunology, plays a crucial role in OM pathogenesis. Inflammatory conditions have been shown to directly affect osteoclast function, leading to imbalanced bone turnover and bone loss. The chronicity of the infection and sustained inflammation play a pivotal role in this panorama. OM progresses through various stages, each characterized by different clinical presentations and biomarker profiles. The acute infection stage is marked by severe pain, fever, and elevated levels of inflammatory markers like CRP and IL-6. The subacute stage is characterized by less pronounced systemic symptoms but persistent localized inflammation. Biomarkers such as sCD14-ST and CCL2 may show significant elevation. Long-standing infections have periods of quiescence and exacerbation. Chronic inflammation leads to the formation of sequestra and extensive bone damage. Biomarkers like SuPAR and the RANKL/OPG ratio are often elevated in this phase. Individual features also have an impact on OM development. Indeed, the incidence and severity of OM may differ between males and females due to hormonal influences on the immune response and bone metabolism. Young and elderly patients are more susceptible to OM due to differences in immune system function and bone remodeling rates. Currently, conditions like diabetes and obesity significantly impact the progression and management of OM. Diabetic patients often have impaired immune responses and poor vascularization, leading to chronic infections and delayed healing. Obesity is associated with systemic inflammation, which can exacerbate the severity of OM. Moreover, immunosuppressive conditions and treatments can predispose individuals to OM and complicate treatment efficacy. Thus, early diagnosis and treatment are crucial for preventing progression from acute to chronic OM. Biomarker levels vary across different stages, providing valuable diagnostic and prognostic information [6,7].

Among serum circulating biomarkers, osteoimmunological markers offer insight into both the bone and immune systems [8]. Recent research has demonstrated a reciprocal interaction between these systems, leading to the emergence of the field of osteoimmunology. This discipline explores how the immune system influences bone turnover and remodeling under both physiological and pathological conditions. Specifically, it has been shown that inflammatory conditions directly impact osteoclast function, causing an imbalance in bone turnover and ultimately resulting in bone loss [9].

Emerging biomarkers such as soluble CD14 subtype (sCD14-ST) have shown promise in diagnosing infections. sCD14-ST is a soluble form of a glycoprotein expressed on monocytes and macrophages, released into circulation in response to pro-inflammatory signals. This study aims to investigate the diagnostic potential of a panel of osteoimmunological and inflammatory serum biomarkers, focusing on sCD14-ST. Recent studies highlight the diagnostic and prognostic potential of sCD14-ST in various infections, including pulmonary infections, COVID-19, and sepsis [10]. Its role in orthopedic infections, such as prosthetic joint infection [11], has been explored, but its application in OM remains under-investigated. This study seeks to fill this gap by assessing the diagnostic and prognostic value of sCD14-ST in OM patients.

The study population included 77 patients undergoing orthopedic surgery, with 42 confirmed OM cases. Infections were predominantly caused by *Staphylococcus aureus*, with some cases involving co-infections with *Staphylococcus agalactiae* and *Enterobacter cloacae*. The immune response to these infections involves a complex interplay between pro-inflammatory and anti-inflammatory signals, influencing bone remodeling and disease progression. OM progresses through various stages, impacting bone tissue differently. Early diagnosis and understanding of biomarker levels at different stages are crucial for effective treatment.

Based on these premises, the present study evaluates the diagnostic and prognostic role of sCD14-ST alongside other markers such as soluble urokinase plasminogen activator receptor (SuPAR), chemokine CCL2, anti-inflammatory cytokine IL-10, Wnt inhibitors DKK-1 and Sclerostin, and the RANKL/OPG ratio. These biomarkers provide a multifaceted approach to diagnosing and understanding the inflammatory and osteoimmunological aspects of OM.

## 2. Materials and Methods

### 2.1. Study Design, Patient Enrollment

Patients with osteomyelitis or mechanical-device-related bone infection were included in the study. Patients were treated at the Reconstructive Surgery of Osteoarticular Infections department of the Galeazzi-Sant’ Ambrogio Hospital in Milan (Italy), from December 2021 to June 2023.

The population of 77 selected patients (42 males, 35 females; mean age 57.35 ±16.93) undergoing orthopedic surgery was enrolled from IRCCS Istituto Ortopedico Galeazzi (Milan) and subdivided into two groups according to the presence of OM: 42 patients with OM at the level of the tibia (14), fibula (7), knee (1), ankle (5), upper limb (4), femur (7), and calcaneus (4); and 35 patients without OM but undergoing an orthopedic surgery for arthrosis.

According to Cierny and Mader’s anatomopathological classification, 18 patients were classified as type 1 osteomyelitis, 2 as type 2, and 22 as type 3. The host type based on comorbidities was classified according to Cierny and Mader’s classification prevalence as type A and type B. All patients underwent evaluation with pre-operative clinical and laboratory tests, X-ray, CT, and MRI scan. Concerning clinical presentation and local inflammatory signs (redness, swelling, pain, local warmth), all patients were considered to have a chronic infection (presence of only one or no signs of local inflammation). A total of 17 patients had a draining sinus at the time of surgery. In OM patients, the infection was also confirmed by a positive culture test, with isolation of the causal agent: the majority of the infection was due to Staphylococcus aureus, while only three patients displayed a coinfection of *Staphylococcus aureus* and *Staphylococcus agalactiae* and only one patient showed a coinfection of *Staphylococcus agalactiae* and *Enterobacter cloacae.*

The inclusion criteria were as follows: age greater than or equal to 18 years; patients referring to the CRIO center of the IRCCS Galeazzi Orthopedic Institute for osteomyelitis or other orthopedic surgery without diagnosis of infection; subjects of both sexes; signature of informed consent The exclusion criteria were as follows: age younger than 18 years; pregnant women, breast-feeding or planning to become pregnant during the study; subjects with uncontrolled diabetes mellitus; subjects who are affected by any congenital or acquired immunodeficiency disease, on chronic corticosteroid therapy, with solid organ or bone marrow transplantation, cytotoxic chemotherapy within the previous 12 months (or planned within the next 12 months); patients with severe illnesses or malignancies; patients with a condition that made it impossible to obtain informed consent (i.e., a physical or mental incapacity, a history of drug and alcohol abuse, patients unlikely to cooperate, or declared legally incompetent); patients unwilling to sign an informed consent or unwilling or unable to come to the follow-up consultations.

Diagnosis of osteomyelitis was performed via:-Routine pre-operative radiological investigations (X-ray and MRI);-Laboratory tests required by clinical routine:

Pre-operative: CRP, complete blood count with leukocyte formula;

First post-operative blood sample (1 post-operative day): CRP, complete blood count with leukocyte count;

Second post-operative blood sample, if performed (3 weeks post-operative +/1 day): CRP, complete blood count with leukocyte count.

The determinations were collected by recording the values of the blood samples taken as part of the checks conducted in the normal diagnostic flow. The study did not involve genetic or diagnostic investigations and any residual samples at the end of the analysis were destroyed at the end of the study.

Safety and evaluation of adverse reactions in relation to the techniques used: Participating subjects were closely monitored for any undesirable effects. Any possible reaction was recorded in the appropriate section of the CRF.

Data collection for withdrawn subjects: Even if subjects could withdraw from the study, side effects would have been recorded and the data collected up to that point were retained. No side effects were registered. Samples not analyzed or residual samples were eliminated.

Risk/benefit ratio: No specific risks associated with participation in this study were identified. Clinical and imaging data were collected only from routinely performed evaluations.

The patients had no direct benefits from participation in the study, but they allowed the collection of data useful for the knowledge and diagnosis of the pathology under consideration.

All the patients underwent pre- and post- (24 h) surgery radiographic tests. In OM patients, post-operative Rx confirmed infected bone removal.

Blood drawing was performed from all patients at T0 (before surgery) and T1 (24 h after surgery). Plasma + EDTA and serum samples were obtained and stored at −20 °C.

The research related to human use complied with all the relevant national regulations and institutional policies and was in accordance with the tenets of the Helsinki Declaration. Written informed consent was obtained from all participants. The study was approved by the local ethics committee (CE of IRCCS San Raffaele Hospital, Milan, CE 146/INT/2021). Details that might disclose the identity of the subjects under the study were omitted, following HIPAA. The study was registered as OSOS L4164 to CLINICALTRIALS.GOV.

### 2.2. Quantification of sCD14-ST, IL-6, and PCT

sCD14-ST concentration (pg/mL), IL-6 concentration (pg/mL), and PCT concentration (ng/mL) were measured using CL-1200i (Mindray, Shenzen, China), according to the manufacturer’s protocol based on the sandwich immunoenzymatic assay (CLIA) using monoclonal antibodies for sCD14-ST, IL-6, and PCT. CL-1200i is a floor-standing, fully automatic, chemiluminescence immunoassay system offering up to 240 tests per hour using the ALP-AMPPD principle.

For the sCD14-ST concentration, in the first step, a sample paramagnetic microparticle coated with monoclonal anti-sCD14-ST antibody (mouse) and monoclonal anti-sCD14-ST antibody (mouse)–alkaline phosphatase conjugate were added into a reaction vessel. After incubation, sCD14-ST present in the sample bound to both the anti-sCD14-ST antibody-coated microparticle and anti-sCD14-ST antibody–alkaline phosphatase-labeled conjugate to form a sandwich complex. The microparticle was magnetically captured while other unbound substances were removed by washing. In the second step, the substrate solution was added to the reaction vessel. It was catalyzed by anti-sCD14-ST antibody (mouse)–alkaline phosphatase conjugate in the immunocomplex retained on the microparticle. The resulting chemiluminescent reaction was measured as relative light units (RLUs) by a photomultiplier built into the system. The amount of sCD14-ST present in the sample is proportional to the light units (RLUs) generated during the reaction. The sCD14-ST concentration can be determined via a calibration curve. The measurement range of the sCD14-ST assay was 20–20,000 pg/mL.

Assay procedures: Before loading the sCD14-ST (CLIA) reagent kit on the analyzer for the first time, the unopened reagent bottle should be inverted gently for at least 30 min to resuspend the microparticles that could have settled. The assay requires 60 uL of sample volume for a single test, not including the dead volume of the sample container. Calibration: The CL series sCD14-ST (CLIA) has been standardized against a commercial sCD14-ST test. The specific information about the calibration master curve of the sCD14-ST (CLIA) kit is stored in the barcode attached to the reagent pack. It is used in combination with sCD14-ST calibrators for the calibration of the specific reagent lot. When performing calibration, the information of the calibration master curve is scanned from the barcode into the system, and then, the sCD14-ST calibrators are used. Quality control: quality control results should be within acceptable ranges. If not, the associated test results are invalid and the sample must be retested. Calculation: the analyzer automatically calculates the analyte concentration of each sample on the calibration master curve read from the barcode, and a 4-parameter logistic curve fitting with the relative light units (RLUs) generated from the calibrators’ defined concentration values, in pg/mL. Limitations: the measuring range of the reagent kit is 57–20,000 pg/mL. For results higher than 20,000 pg/mL, a 1:5 dilution is recommended.

Performance characteristics: Limit of blank (LoB) = 10 pg/mL; limit of detection (LoD) = 20 pg/mL; limit of quantification (LoQ) = 57 pg/mL. Precision was determined following the Clinical & Laboratory Standards Institute (CLSI) protocols for EP5-A3. Two levels of quality controls and two levels of samples were tested, with 2 runs each day, in duplicate, for a total of 20 days (*n* = 80). The total precision was <8%. Linearity was determined following the Clinical & Laboratory Standards Institute (CLSI) protocols for EP5-A3. A high-concentration sCD14-ST sample (>20,000 pg/mL) was mixed with a low-concentration sample (<20 pg/mL) at different ratios, generating a series of solutions. The linearity was demonstrated in this range, with a correlation coefficient r > 9.900.

Analytical specificity: Samples with hyperlipidemia, icterus, hemolysis, high total protein, high rheumatoid factor, or positive antinuclear antibody (ANA) may cause inaccurate results. Mindray sCD14-ST contains anti-interference ingredients, which can reduce the impact of HAMA in samples. However, some interference may still exist, as in the case of patients with severe renal disease showing higher concentrations of sCD14-ST. For this reason, patients with severe renal disease were excluded from this study.

On the same sample, CL-1200i can also measure IL-6 and PCT concentrations, according to the same manufacturer’s protocol based on the sandwich immunoenzymatic assay (CLIA). The measurement ranges of the IL-6 and PCT assays were 1.5–5000 pg/mL and 0.02–100 ng/mL. respectively.

The CL series interleukin-6 (IL-6) assay is a chemiluminescent sandwich immunoassay (CLIA) for the quantitative detection of IL-6 in human serum and plasma. This assay can be used to monitor the immune status and inflammatory reaction. In the first step, sample diluent and paramagnetic microparticles are coated with IL-6 antibody and anti-IL-6. Antibody–alkaline phosphatase (ALP) conjugates are added into the reaction cuvette. After incubation, Il-6 in the sample will bind to IL-6-antibody-coated microparticles and anti-IL-6 antibody–alkaline phosphatase (ALP) conjugates. Afterward, microparticles are magnetically captured while other unbound substances are removed by washing. In the second step, a chemiluminescent substrate (sodium 3-(1R3S5r7r)-4 methoxyspriroladamentane-2–3-(1,2) dioxetane-4-y phenyl phosphate AMPPD) is added to the rection cuvette. AMPPD removed a phosphate group by ALP, forming an unstable intermediate structure, which produces the methyl methoxybenzoate anion through intramolecular electron transfer. Chemiluminescence is generated when the methyl methoxy benzoate anion in the excited state returns to the ground state. The resulting chemiluminescent reaction is measured as relative light units (RLUs) by a photomultiplier in the instrument. The amount of IL-6 present in the sample is proportional to the RLUs generated during the reaction.

The IL-6 results can be determined by a calibration curve, which is established on an encoded calibration master curve and three level product calibrators. The analyzer automatically calculates the analyte concentration of each sample on the calibration master curve read from the barcode, and a 4-parameter logistic curve fitting with the relative light units (RLUs) generated from the calibrators’ defined concentration values, in pg /mL. Performance assay characteristics: lower limits of measurements: LoB = 1.0 pg/mL; LoD = 1.5 pg/mL; LoQ = 2.5 pg/mL; linearity: 1.5−5000 pg/mL; within-run precision ≤ 5%; total imprecision ≤ 8%; high-dose hook = 200,000 pg/mL; time to first results = 16 min.

The CL series Procalcitonin assay is a chemiluminescent immunoassay (CLIA) for the quantification of procalcitonin (PCT) in human serum or plasma.

The CL series PCT assay is a two-site immunoenzymatic assay to determine the level of procalcitonin. In the first step, the sample, parametric microparticles coated with monoclonal anti-PCT antibody (mouse) and monoclonal anti-PCT antibody 8 (mouse)–alkaline phosphatase conjugates are added into a reaction cuvette. After incubation, PCT present in the sample binds to both anti-PCT antibody-coated microparticles and anti-PCT antibody–alkaline phosphatase-labeled conjugate to form a sandwich complex. Microparticles are magnetically captured while other unbound substances are removed by washing. In the second step, the substrate solution is added to the reaction cuvette. It is catalyzed by anti-PCT antibody (mouse)–alkaline phosphatase conjugated in the immunocomplex retained by microparticles. The resulting chemiluminescent reaction is measured as relative light units (RLUs) by a photomultiplier in the instrument. The amount of PCT present in the sample is proportional to the RLUs generated during the reaction. The PCT concentration can be measured via a calibration curve.

Performance assay characteristics: lower limits of measurements: LoB < 0.02 ng /mL; linearity: 0.02–100 ng/mL, with a correlation coefficient r > 0.990; high-dose hook = 1000 ng/mL; time to first results = 16 min. Precision was determined by following the National Committee for Clinical Laboratory Standards’ (NCCLS’) protocols EP5-A2.: within-run precision = 6.10%; total imprecision = 6.16%.

### 2.3. Quantification of Osteommunological Markers RANKL, OPG, DKK-1, and SOST

RANKL, OPG, and SOST were measured using an ELISA sandwich Quantikine Assay, according to the manufacturer’s protocol (Pikokine TM ELISA for OPG, quantitative sandwich ELISA for RANKL and SOST, MyBioSource, San Diego, CA, USA). DKK-1 was measured using an ELISA sandwich Quantikine Assay, according to the manufacturer’s protocol (R&D System, Minneapolis, MN, USA).

### 2.4. Quantification of Inflammatory Markers CRP, TNFa, IL-10, CCL2, and SuPAR

CRP was measured using immunoturbidimetry on an automated biochemical analyzer (Olympus CRP-Latex assay, Central Valley, PA, CA, USA).

SuPAR was measured by SuPARnostic ELISA Assay, according to the manufacturer’s protocol (Virogates, Denmark).

CCL2, TNFa, IL-10 were measured using an ELISA sandwich Quantikine Assay, according to the manufacturer’s protocol (R&D System, Minneapolis, MN, USA).

### 2.5. Statistical Analysis

The normality of the distribution of the groups was verified by KS normality for all the biomarkers analyzed. The Kolmogorov–Smirnov test is used to test the null hypothesis that a set of data comes from a normal distribution. The normality KS test was performed using the following null and alternative hypotheses: H0: The data are normally distributed; HA: The data are not normally distributed. For each parameter, the maximum value was not greater than this critical value, so we failed to reject the null hypothesis. This means we can assume that our sample data are normally distributed.

Results are expressed as the mean standard deviation (SD). Statistical analysis was performed by means of the one-way ANOVA test, considering *p* < 0.05 significant and *p* < 0.005 very significant. The PRISM 5.0 software was used for correlation analysis. The correlation between values measured by the different assays was evaluated by the Pearson correlation coefficient (r2). Linear regression analysis was evaluated between the different groups of data, measuring the 95% confidence interval of the regression line. Statistical analysis of the receiver operating characteristic (ROC) curves and the area under the curve (AUC) was performed by the PRISM 5.0 software. ROC analysis provides tools to select possible optimal models and to discard suboptimal ones independently from (and before specifying) the cost context or the class distribution. ROC analysis is related directly and naturally to the cost/benefit analysis of diagnostic decision making. The AUC is the result of the integration of all of the points along the path of the curve and simultaneously computes sensitivity and specificity, offering an estimator of the overall behavior and accuracy of a test. The area under the curve (often referred to as simply the AUC) is equal to the probability that a classifier will rank a randomly chosen positive instance higher than a randomly chosen negative one (assuming ‘positive’ ranks higher than ‘negative’). ROC AUC varies between 0 and 1. The AUC offers an estimation of the probability of correctly classifying a random subject (test accuracy); an AUC of 0.7 indicates a 70% likelihood of correctly classifying the case. In general, AUC values are interpreted as follows: 0.5–0.6 (failed), 0.6–0.7 (worthless), 0.7–0.8 (poor), 0.8–0.9 (good), >0.9 (excellent). A value of 0 represents chance performance whilst 1 represents perfect performance.

To exclude potential confounders, the two groups of patients were sex- and age-matched. For the same reason, the juvenile form of OM was excluded from this study. To reduce analytical test variability, all the samples were evaluated under the same experimental conditions, namely, in the same test all at once, for each parameter.

## 3. Results

### 3.1. sCD14-ST and Inflammatory Biomarkers in OM Patients: Serum Levels and ROC Curve

Figure 1 shows the sCD14-ST evaluation in OM patients (dark gray bars) and the control group of OM-negative patients (light gray bars), along with the associated biomarker panel provided by the Mindray CL-1200i sandwich immunoenzymatic assay (CLIA). This panel analyzes, in the same serum sample, the inflammatory biomarkers procalcitonin, the inflammatory primary cytokine IL-6, and sCD14-ST, to provide a more comprehensive image of the inflammatory response to the infection. As shown in Figure 1a, sCD14-ST displayed a very significant difference between OM-positive and -negative patients at T0, which remains quite significant also at T1, 2 hours post-surgery. This strong ability to distinguish OM-positive and -negative patients is confirmed by a very high sCD14-ST ROC AUC (0.978) (Figure 1b).

In the longitudinal evaluation, sCD14-ST displayed a very significative decrease in OM patients from T0 to T1, while in OM-negative patients it remained nearly stable.

IL-6, as shown in Figure 1c, displayed a significantly higher level in OM patients compared to controls, and this difference remained stable at T1. The diagnostic power in OM patients of IL-6 is confirmed by a good ROC AUC value (0.737), shown in Figure 1d.

On the contrary, Procalcitonin (Figure 1e) displayed very low levels in both OM-positive and -negative patients, showing no difference between the two groups, either at T0 or T1. Similarly, C-reactive protein (CRP) displayed a very low level at T0, with no significant difference between OM-positive and -negative patients (Figure 1g). CRP showed a very small but not significant decrease at T1, but maintained no significant difference between the groups. In both the biomarkers, the AUC ROC confirmed the absence of diagnostic accuracy in OM patients, with values of 0.556 (Figure 1f) and 0.592 (Figure 1h), respectively.

### 3.2. Infection and Inflammatory Markers in OM Patients: Serum Levels and ROC Curve

Figure 2 shows the serum level of different inflammatory markers in OM patients (dark gray bars) and OM-negative patients (light gray bars).

The inflammatory cytokine TNF-alpha (Figure 2a) displayed a small but weakly significant increase in OM patients compared to OM-negative ones, as confirmed by a low level of the ROC AUC value (AUC = 0.647, Figure 2b). TNF-alpha showed a weak but significant decrease in the longitudinal evaluation, reaching a comparable level to OM-negative patients at T1.

The inflammatory chemokine CCL2 (Figure 2c) displayed a strong and highly significant increase in OM patients, as confirmed by a very good AUC ROC (AUC = 0.945, Figure 2d). It also showed a significant decrease in the longitudinal evaluation from T0 to T1 in OM patients, while it showed a small decrease or none at all in OM-negative patients.

Conversely, the anti-inflammatory cytokine IL-10 (Figure 2e) showed a significantly higher level at T0 in OM-negative patients compared to OM-positive ones. These values remained stable at T1 for both groups, maintaining statistically significant differences between the two groups of patients, as confirmed by a good AUC ROC (AUC = 0.908, Figure 2f).

The infection biomarker SuPAR (Figure 2g) showed a strong and very significant increase in OM patients compared to OM-negative ones, as confirmed by a very good AUC ROC (AUC = 0.967). In the longitudinal evaluation at T1, SuPAR displayed a significant decrease in OM-positive patients and a very small but not significant decrease in OM-negative patients, thus maintaining a very statistically significant difference between these two groups.

### 3.3. Osteoimmunological Markers in OM Patients: Serum Levels and ROC Curve

The serum levels of the main osteoimmunological biomarkers are shown in Figure 3.

RANKL, a marker of bone resorption, displayed a strong increase in OM patients, but this difference was not statistically significant at T0 due to the wide standard deviation. On the contrary, at T1, though there was an evident decrease in OM patients, the difference between OM-positive and -negative patients was significant, with a higher level of RANKL in OM patients. This behavior is confirmed by a weak ROC AUC (AUC = 0.794), as shown in Figure 3b.

Conversely, the osteoprotective marker OPG (Figure 3c) showed a very significant increase in OM-negative patients compared to OM-positive ones at T0, as confirmed by a good AUC ROC (AUC = 0.872). These levels were maintained stable at T1, confirming a strong significant difference between the two groups.

In order to obtain a more informative result, the ratio RANKL/OPG (Figure 3e) showed a more significant difference between OM-positive and -negative patients compared to RANKL or OPG alone. In particular, the ratio RANKL/OPG displayed a very significantly higher level in OM-positive patients compared to negative ones, as confirmed by a very good value of ROC AUC (AUC = 0.938, Figure 3f), though there was a quite high standard deviation in OM patients due to the high standard deviation of RANKL. This difference was maintained at T1, and even though the ratio displayed a little decrease in OM patients, this difference was not significant in the longitudinal evaluation.

The panel of osteoimmunological biomarkers also included the two Wnt pathway inhibitors: Sclerostin, encoded by the gene SOST, and DKK-1.

Figure 3g, described the serum level of DKK-1, which displayed a very significant increase in OM-positive patients, and this difference remained stable also at T1. These results are confirmed by the good AUC ROC (AUC = 0.917), shown in Figure 3h.

Sclerostin (SOST, Figure 3i) showed a very similar behavior to DKK-1, with a very significantly higher level in OM-positive patients compared to OM-negative ones, even with a lower AUC ROC (AUC = 0.819). Similarly, the levels of SOST remained stable in both OM-positive and -negative patients at T1.

### 3.4. sCD14-ST Correlation with Inflammatory and Osteoimmunological Biomarkers

To evaluate the potential role of sCD14-ST as a diagnostic tool to compose a panel of diagnostic biomarkers of OM, the correlation between sCD14-ST and all the inflammatory and osteoimmunological biomarkers measured in this study was evaluated, as shown in Figure 4.

sCD14-ST was first correlated with the primary inflammatory mediators evaluated, IL-6 (Figure 4a) and TNF-alpha (Figure 4d), resulting in a good correlation with IL-6 (r2 = 0.915) and a lower correlation with TNF-alpha (r2 = 0.906). In agreement with these results, sCD14-ST showed a negative correlation with the anti-inflammatory cytokine IL-10 (r2 = 0.870, Figure 4c), while it showed a positive correlation with the inflammatory chemokine CCL2 (r2 = 0.961, Figure 4d).

The infection biomarker SuPAR (Figure 4e) showed a good positive correlation with sCD14-ST (r2 = 0.823), confirming the role of this molecule in the diagnosis of infection.

sCD14-ST was then correlated more specifically with the osteoimmunological biomarkers evaluated.

In agreement with the serum levels in OM-positive and -negative patients, OPG (Figure 4f) showed a negative correlation with sCD14-ST (r2 = 0.933), while RANKL (Figure 4g) showed a positive but weaker correlation with sCD14-ST (r2 = 0.752). A better positive correlation with sCD14-ST was obtained considering the ratio RANKL/OPG (Figure 4h, r2 = 0.944). Consistently with the serum level in OM patients, the other two osteoimmunological biomarkers, DKK-1 (Figure 4i) and SOST (Figure 4l), showed a similar positive correlation with sCD14-ST (r2 = 0.892 and r2 = 0.925, respectively).

### 3.5. Surgical and Clinical Treatment

Once the diagnosis was made, the indication was given for elective surgical treatment of sequestrectomy and debridement of the osteomyelitis focus. The surgery performed can be summarized as follows: skin incision targeted to the pathologic portion of bone, skeletonization of the bone, the opening of a bone window with an appropriate saw to access the medullary canal, and subsequent thorough cleaning of the canal with the removal of any bone sequestration. Figure 5 reports examples of pre-operative and post-operative radiographic and MRI images of chronic osteomyelitis of the distal tibia: the post-operative images confirm the recovery of bone tissue compared to the pre-operative condition. Multiple tissue samples were collected in all patients for culture and histological examination.

All patients received antibiotic therapy for 4–6 weeks (at least 2 of which were intravenous) starting from the day of surgery. Drugs were chosen in consultation with the infectious disease specialist based on prior isolations (if available), and subsequently, modified if necessary, in light of intraoperative culture results. Healing was assessed, upon discontinuation of antibiotic therapy, based on clinical course and normalization of inflammatory indices. No patients experienced recurrence at the current minimum follow-up of 6 months (maximum 2 years). The control group enrolled orthopedic patients with no history of osteoarticular infection who were candidates for elective surgery for first-implant shoulder, hip, or knee prostheses or arthroscopic shoulder or knee surgery.

## 4. Discussion

Osteomyelitis, or inflammation of bone, is most commonly caused by invasion of bacterial pathogens into the skeleton. Bacterial osteomyelitis is difficult to treat due to widespread antimicrobial resistance in the preeminent etiologic agent, the Gram-positive bacterium *Staphylococcus aureus*. Bacterial osteomyelitis triggers pathological bone remodeling, which in turn leads to sequestration of infectious foci from innate immune effectors and systemically delivered antimicrobials. Treatment of osteomyelitis, therefore, typically consists of long courses of antibiotics associated with surgical debridement of necrotic infected tissues.

Diagnosis of osteomyelitis is one of the most challenging tasks in orthopedics. Delays or diagnostic errors can lead to inappropriate treatment decisions and potentially adverse outcomes for the patient. Accurate diagnostic tests are crucial for enabling surgeons to perform interventions at the optimal time and in the most effective manner.

Among infection biomarkers, an emerging molecule is the soluble CD14 subtype (sCD14-ST), a soluble form of a glycoprotein expressed on the membrane of monocytes and macrophages, released into circulation upon a pro-inflammatory signal against an infectious agent [12,13]. Circulating levels of sCD14-ST have been shown to increase in response to various infections, correlating with disease severity. Notably, serum levels of sCD14-ST are good biomarkers for infections following trauma and invasive surgical procedures. The significant role of sCD14-ST in orthopedic infections is particularly evident in complex and challenging diagnoses, such as prosthetic joint infection [14]. For this reason, sCD14-ST could be a promising candidate for the diagnosis of OM, but its role in OM diagnosis has not been investigated yet. Therefore, to our knowledge, this is the first study evaluating the role of serum sCD14-ST in the diagnosis and early prognosis of OM. The study results clearly show a significantly higher level of sCD14-ST in OM-positive patients compared to the control group of patients undergoing orthopedic surgery without infection. The choice of this control group over a healthy population ensures that both groups share similar surgical conditions, which could elicit a basal inflammation response to the surgical procedure, making the presence of infection leading to OM the only distinguishing factor. This approach highlights the diagnostic power of the biomarkers evaluated in the differential diagnosis of OM. These results indicate a strong diagnostic value, as confirmed by the good ROC curve AUC value, aligning with previous evidence suggesting an important role for sCD14-ST in different orthopedic infections, such as post-operative spinal infection [14], hip and knee replacements [15], septic arthritis [16], and odontogenic infections [17].

The study also aimed to perform a longitudinal evaluation, at least at the early 24 h post-surgery mark, to observe the short-term variations in biomarkers. The significant decrease in sCD14-ST shortly after surgery suggests its potential for longitudinal monitoring of OM patients during post-surgical follow-up, which is crucial given the high recurrence rate in OM [18].

A standard approach to diagnosing osteomyelitis is based on clinical suspicion. Initial evaluation includes measurement of serum C-reactive protein (CRP) and the erythrocyte sedimentation rate (ESR), radiographs, and a blood culture [19].

sCD14-ST was found to be particularly significant when compared to traditional biomarkers in the panel, such as IL-6, Procalcitonin (PCT), and C-reactive protein (CRP).

To create a panel of biomarkers, the study initially focused on the two primary inflammation biomarkers commonly used in clinical settings, PCT and CRP, which are well-documented as the main circulating biomarkers of inflammation [20].

Elevated ESR and CRP levels are nonspecific for osteomyelitis because many different inflammatory states will have elevated ESR and CRP levels; a patient with acute osteomyelitis usually has a normal ESR and CRP level, while they can decrease in chronic OM. Our patients present chronic OM caused by arthrosis, so is not surprising that canonical inflammatory markers such as CRP and PCT were not significantly elevated in our OM patients compared to not-infected patients.

PCT showed no variation between OM-positive and -negative patients in our study, possibly due to the uniformly low levels in both groups, falling below the clinical threshold. This might obscure any differing responses between OM patients. Recent studies have suggested that the diagnostic accuracy of PCT in OM increases with higher cut-off levels [21]. Consequently, the lack of diagnostic significance in our study, as confirmed by the AUC ROC, may stem from these notably low levels. Similarly, CRP demonstrates very low levels and diagnostic accuracy. Although widely used in clinical practice for diagnosing inflammation and infection, increasing evidence suggests the limitations of this marker in differentiating OM. These limitations are evident across different cut-off values, patient subsets, and cases of low pathogen virulence, where CRP levels can be minimal [22]. Indeed, CRP alone is not always effective in distinguishing between osteomyelitis and degenerative spine diseases [23]. Moreover, CRP’s usefulness in this study is limited due to the nonspecific increase caused by surgery. This highlights that a combination of biomarkers, rather than a single marker, could provide more informative results. For this reason, additional biomarkers were evaluated in this study. The primary inflammatory cytokine, IL-6, acts as a pleiotropic cytokine, with multiple roles in immune response, tissue repair, and regeneration. The rapid production of IL-6 contributes in host defense during tissue injury and infection, whereas excessive production and accumulation of IL-6 often lead to disease pathology. IL-6 has previously been recognized as important in orthopedic infections, such as prosthetic joint infection [24], and recent evidence suggests its role in the diagnosis of OM [25].

Consistent with this evidence, IL-6 displays a significant increase in OM patients, suggesting that it could be a potential prime candidate for inclusion in a biomarker panel for OM diagnosis. However, serum levels of IL-6 remained unchanged at early longitudinal time points, indicating that it requires more time to decrease post-surgery, making it unsuitable for early longitudinal evaluation. This is also confirmed by an ROC AUC value of 0.737.

In recent studies on OM diagnosis, IL-6 was studied alongside TNF-alpha, another primary inflammatory cytokine [26]. The findings suggest that the pre-operative levels of these two molecules combined could be a reliable biomarker for chronic osteomyelitis in extremities. In our OM patients, TNF-alpha displayed a weak increase during the pre-operative evaluation, suggesting weak diagnostic accuracy, as confirmed by a relatively low AUC ROC. However, TNF-alpha exhibited a significant decrease in the early longitudinal evaluation, indicating that it may be more useful as a prognostic biomarker rather than a diagnostic one for OM. A recent comprehensive study evaluating the secretome in *Staphylococcus aureus*-induced OM [27] described the role of not only primary inflammatory cytokines but also different chemokines. Among these, the present study focused on the chemokine CCL2, the principal key mediator of monocyte recruitment, which triggers the release of sCD14-ST into circulation [28]. Our findings showed a significant increase in CCL2 levels in OM patients, indicating a good diagnostic accuracy with a high ROC AUC. Additionally, there was a notable decrease in CCL2 levels shortly after surgery, suggesting its potential as both a diagnostic and prognostic biomarker for OM. These results align with previous evidence that osteoblasts can produce CCL2 in *Staphylococcus aureus*-induced OM in both murine models and human bone tissue [29]. In addition, CCL2 is known to promote chemotactic recruitment, RANKL production, and osteoclast activity in response to inflammatory cytokine stimuli [30], confirming its role in osteoimmunological alterations leading to bone erosion in OM.

Since the inflammatory response is a self-regulating mechanism involving both pro- and anti-inflammatory stimuli, this study also evaluated the anti-inflammatory cytokine IL-10 to complete the panel of inflammatory biomarkers. In OM, it has been demonstrated that an increased number of monocytes and macrophages induce the production of TNF-alpha while reducing the production of IL-10, thereby shifting the TNF-alpha/IL-10 ratio towards an inflammatory response [31]. This imbalance directly impacts the osteoimmunological system, specifically RANK/RANKL/OPG, as TNF-alpha stimulates RANKL expression while IL-10 reduces osteoclastogenesis [32]. Therefore, the reduction in IL-10 in OM patients may promote bone loss by directly regulating the effect of RANKL. In the case of low levels of CRP, probably due to low pathogen virulence, specific parameters are needed for diagnosis and longitudinal evaluation [32]. The urokinase plasminogen activator receptor (SuPAR) is the soluble form of uPAR, expressed during inflammation and infection, and detectable in blood, urine, and cerebrospinal fluid. The circulating level of SuPAR provides both qualitative and quantitative information about the severity of infection. It has been described as a primary marker in the differential diagnosis of OM and degenerative spine disease [33], as well as prosthetic joint infection [34]. In this study, significantly higher SuPAR levels in OM-positive patients compared to the control group suggest excellent diagnostic accuracy, as confirmed by a high ROC AUC. Moreover, at T1, although SuPAR levels remained significantly higher in OM patients compared to controls, there was a significant decrease shortly after surgery. This suggests SuPAR could also play a promising role in the longitudinal evaluation of OM, at least in the early post-operative period.

One of the primary causes of OM is *Staphylococcus aureus* infection, but the interaction between infection, immune response, and bone remodeling remains poorly understood. Bone metabolism is strictly linked to the immune system through molecules that act as interfaces between bone and the immune system, representing the basis of osteoimmunology [35]. This interaction is based on the axis RANKL/RANK/OPG system: OPG is a soluble decoy receptor for RANKL, preventing its binding to RANK, and thus, inhibiting osteoclastogenesis. The balance of this axis is critical: increased RANKL leads to osteoclast-mediated bone resorption, while OPG plays a bone-protective role. The RANKL/OPG ratio has been defined as a key determinant of osteoclast-mediated bone loss [36]. Few studies have described the role of the RANK/RAKL/OPG system in OM [37], and there is no evidence yet of a diagnostic application of these molecules in a comprehensive biomarker panel for OM diagnosis and early prognosis. For this reason, our study evaluated serum levels of RANKL and OPG and the RANKL/OPG ratio in OM patients and controls. RANKL levels showed an increase in OM patients, though not statistically significant, aligning with recent evidences that RANKL alone is not a reliable marker to monitor bone destruction in a mouse model of osteomyelitis [38], as confirmed by a low AUC ROC. Nevertheless, post-surgery RANKL levels displayed a small but non-significant decrease, reaching a significant difference compared to controls, suggesting its better utility in longitudinal evaluation rather than the pre-operative diagnosis. Conversely, OPG levels significantly increased in OM-negative patients, as expected by its bone protective role and confirmed by a high AUC ROC. In both OM-positive and OM-negative patients, OPG levels remained stable post-surgery, indicating its suitability for initial diagnosis rather than early longitudinal evaluation. Combining the serum levels of these two molecules into the RANKL/OPG ratio more striking results were obtained. This ratio, previously indicated as a marker of altered bone turnover balance, showed a significant increase in OM patients compared to controls, demonstrating strong diagnostic power, as confirmed by a high AUC ROC. This finding highlights the value of the RANKL/OPG ratio in OM diagnosis over RANKL and OPG individual measurements. In longitudinal evaluation, the RANKL/OPG ratio remained low and stable in OM-negative patients but showed a small, non-significant decrease post-surgery in OM patients, suggesting its use is more appropriate for pre-operative diagnosis rather than post-operative monitoring.

It is described that bone turnover decreases with age, shifting the balance towards bone loss. This process also affects osteoimmunological factors. Inclusion criteria excluded pediatric patients, who can present specific characteristics of OM, in order to consider only adult characteristics.

A recent study reported reference intervals for serum RANKL, OPG, and the RANKL/OPG ratio based on menopausal status and age. It found that median RANKL and RANKL/OPG were higher, while OPG levels were lower in pre-menopausal women compared to post-menopausal women [39]. Also, host response is known to decrease with age according to changes in immunological function, termed immunosenescence. This includes a diminished immune response against infection, increased levels of pro-inflammatory mediators, and a higher risk of autoimmunity. Immunoscenescence is accompanied by low-grade aseptic inflammation, commonly observed in elderly people, defined as inflammaging. This process affects osteoimmunological factors by increasing RANKL levels relative to OPG levels, thus shifting the balance towards bone loss. To exclude age or sex confounding factors, both study groups (OM positive and negative) were age- and sex-matched.

The RANK–RANKL–OPG system and RANKL/OPG ratio are strictly connected to the two other osteoimmunological makers, Sclerostin (SOST) and DKK-1, which are inhibitors of the Wnt pathway. The Wnt pathway promotes bone formation [38], and its inhibition by SOST or DKK-1 leads to bone loss. Sclerostin has been previously associated with bone loss in conditions related to immobilization [40], and anti-Sclerostin therapy has been proposed for osteoporosis or other bone diseases characterized by bone loss [41]. However, little evidence has investigated the role of this molecule in the diagnosis of OM. In our study, SOST displayed a significant increase in OM patients and demonstrated good diagnostic accuracy, as confirmed by a good ROC AUC. This result agrees with previous evidence linking the Wnt pathway to OPG expression, where Wnt pathway activation inhibits bone resorption by osteoblasts and induces the expression of OPG, a bone-protective osteoimmunological molecule. Therefore, inhibition of the Wnt pathway by increase Sclerostin decreases OPG levels, shifting the RANKL/OPG ratio towards bone resorption [42]. Consistently, in this study, the significant increase in SOST in OM patients corresponded to a strong decrease in OPG. Several factors, particularly inflammatory cytokines like IL-6, regulate the expression of Sclerostin by osteocytes [43]. Accordingly, the strong increase in IL-6 observed in our OM patients could be responsible for the increased Sclerostin expression in these patients.

The other Wnt inhibitor evaluated, DKK-1, had a similar trend to Sclerostin, showing a significant increase in OM patients and good diagnostic accuracy confirmed by a high AUC ROC. Several studies have described the link between bone loss induced by reduced Wnt pathway activity and increased expression of DKK-1.

Both Sclerostin and DKK-1 serum levels remained stable post-surgery, indicating that these biomarkers need a longer time to decrease after surgery and are thus more suitable for diagnostic rather than prognostic use in the early post-operative period.

One of the main aims of this study was to highlight the emerging role of sCD14-ST as a new tool for OM diagnosis, becoming the basis for a panel of immunological and osteoimmunological biomarkers. To this purpose, a linear regression analysis was performed to compare sCD14-ST with all biomarkers that were significantly different between OM-positive and -negative patients. Therefore, CRP and PCT, which showed very poor diagnostic accuracy in this study, were excluded from the linear regression evaluation.

Consistent with their diagnostic accuracy, the strongest positive correlation with sCD14 ST was observed for the RANKL/OPG ratio, SuPAR, Sclerostin, and CCL2, while a significant negative correlation was observed for those biomarkers with bone-protective roles, such as OPG, or anti-inflammatory functions, such as IL-10, which were higher in OM-negative patients. A slightly lower correlation with sCD14-ST was observed for RANKL, consistent with its lower diagnostic value compared to other biomarkers evaluated.

The limitation of this study is the short period of longitudinal evaluation, conducted only at a close time point after surgery according to clinical protocols during patients’ hospitalization. Biomarkers that displayed a faster response to infection eradication showed a decrease in OM patients even at this short post-surgical time point. However, a longer longitudinal evaluation with additional time points could provide more information about the prognostic potential of biomarkers that remained stable shortly after surgery.

## 5. Conclusions

In conclusion, these results, for the first time to our knowledge, indicate the diagnostic and early prognostic role of sCD14-ST in OM patients, suggesting that this biomarker could help overcome the limitations of current laboratory tests, such as canonical inflammatory markers, in OM diagnostic approaches. To this purpose, sCD14-ST was evaluated in correlation with a panel of serum biomarkers, providing new insights into not only the inflammatory condition of OM patients but also the direct effect of inflammation on bone remodeling through osteoimmunological molecules. Among biomarkers, a relevant diagnostic role emerged for SuPAR, the chemokine CCL2, the anti-inflammatory cytokine IL-10, and the Wnt inhibitors DKK-1 and Sclerostin, with a striking diagnostic value confirmed for the RANKL/OPG ratio. Some of these biomarkers, namely, CCL2, and SuPAR, also showed prognostic potential in the short post-surgical period.

## Figures and Tables

**Figure 1 diagnostics-14-01588-f001:**
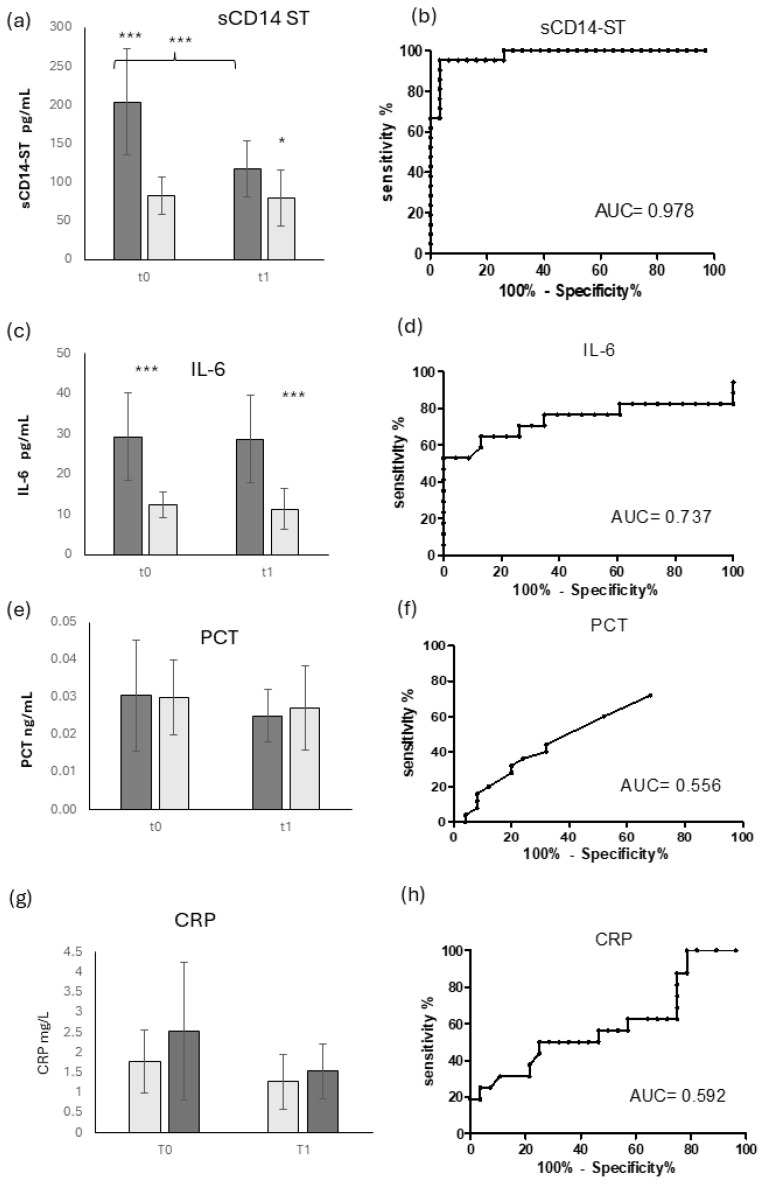
sCD14-ST and inflammatory biomarkers in OM patients: serum levels and ROC curves. Longitudinal evaluation of sCD14-ST (**a**) and inflammatory markers IL-6 (**c**), PCT (**e**), and CRP (**g**) and their ROC curves (**b**,**d**,**f**,**h**), respectively) in OM patients (dark gray) and control patients (light gray). * = *p* < 0.05, quite significant; *** = *p* < 0.001, extremely significant.

**Figure 2 diagnostics-14-01588-f002:**
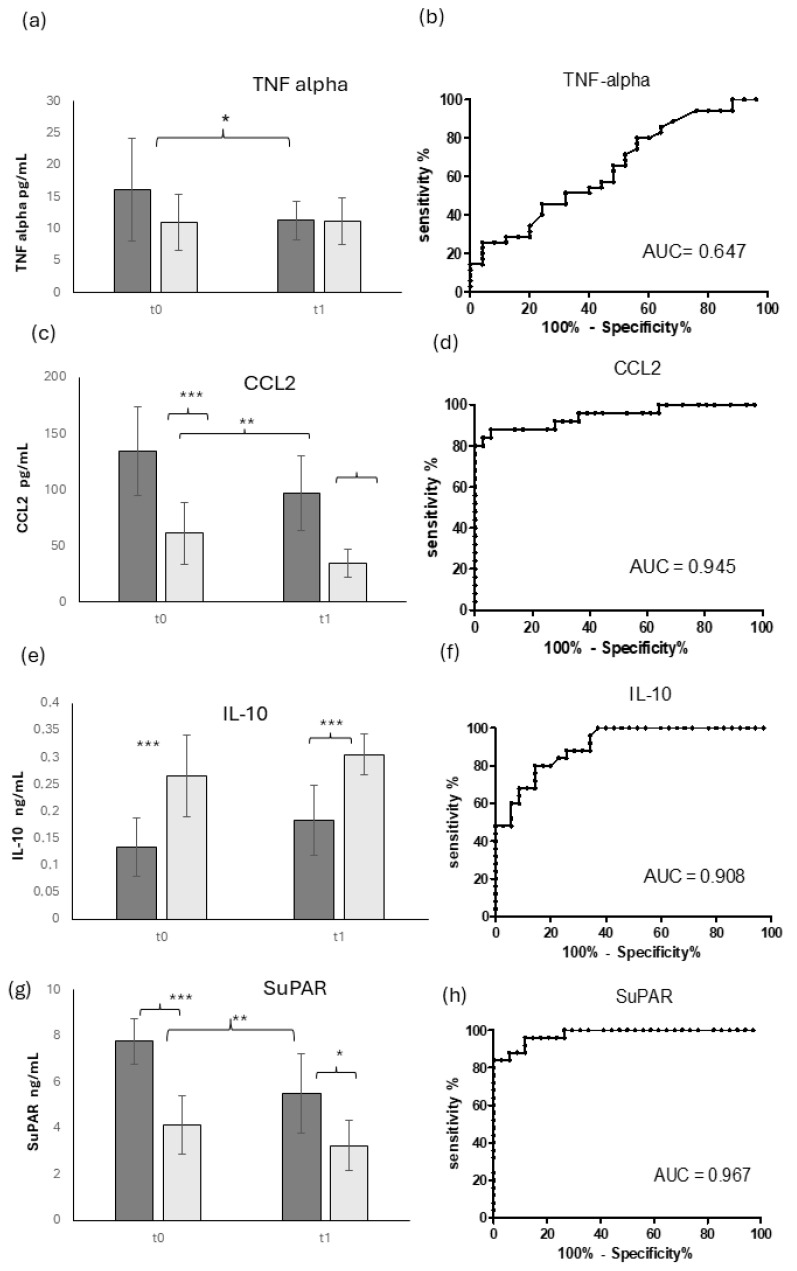
Infection and inflammatory markers in OM patients: serum levels and ROC curves. Longitudinal evaluation of TNF-alpha (**a**), CCL2 (**c**), IL-10 (**e**), and SuPAR (**g**) and their ROC curves (**b**,**d**,**f**,**h**), respectively) in OM patients (dark gray) and control patients (light gray). * = *p* < 0.05, quite significant; ** = *p* < 0.01, very significant; *** = *p* < 0.001, extremely significant.

**Figure 3 diagnostics-14-01588-f003:**
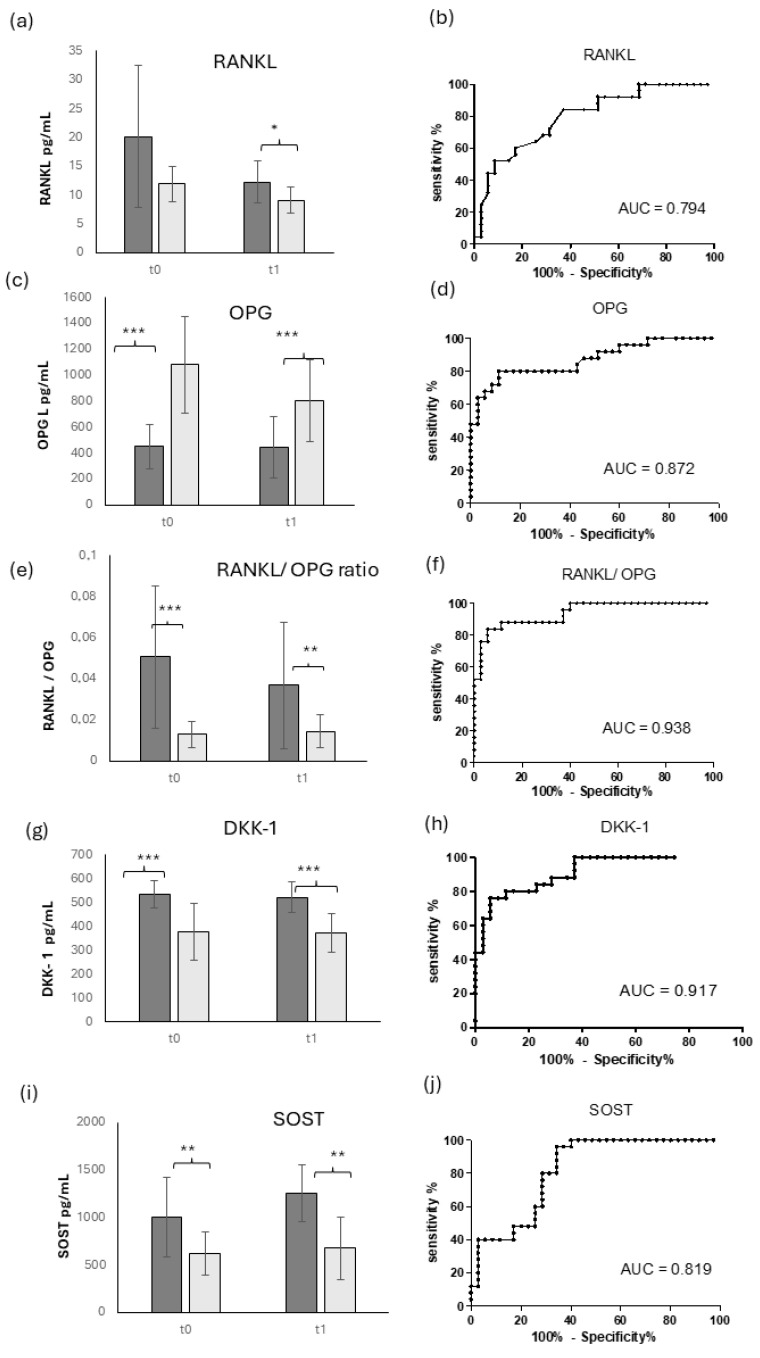
Osteoimmunological markers in OM patients: serum levels and ROC curve. Longitudinal evaluation of RANKL (**a**), OPG (**c**), RANKL/OPG ratio (**e**), DKK-1 (**g**), and SOST (**i**) and their ROC curves (**b**,**d**,**f**,**h**,**j**), respectively) in OM patients (dark gray) and control patients (light gray). * = *p* < 0.05, quite significant; ** = *p* < 0.01, very significant; *** = *p* < 0.001, extremely significant.

**Figure 4 diagnostics-14-01588-f004:**
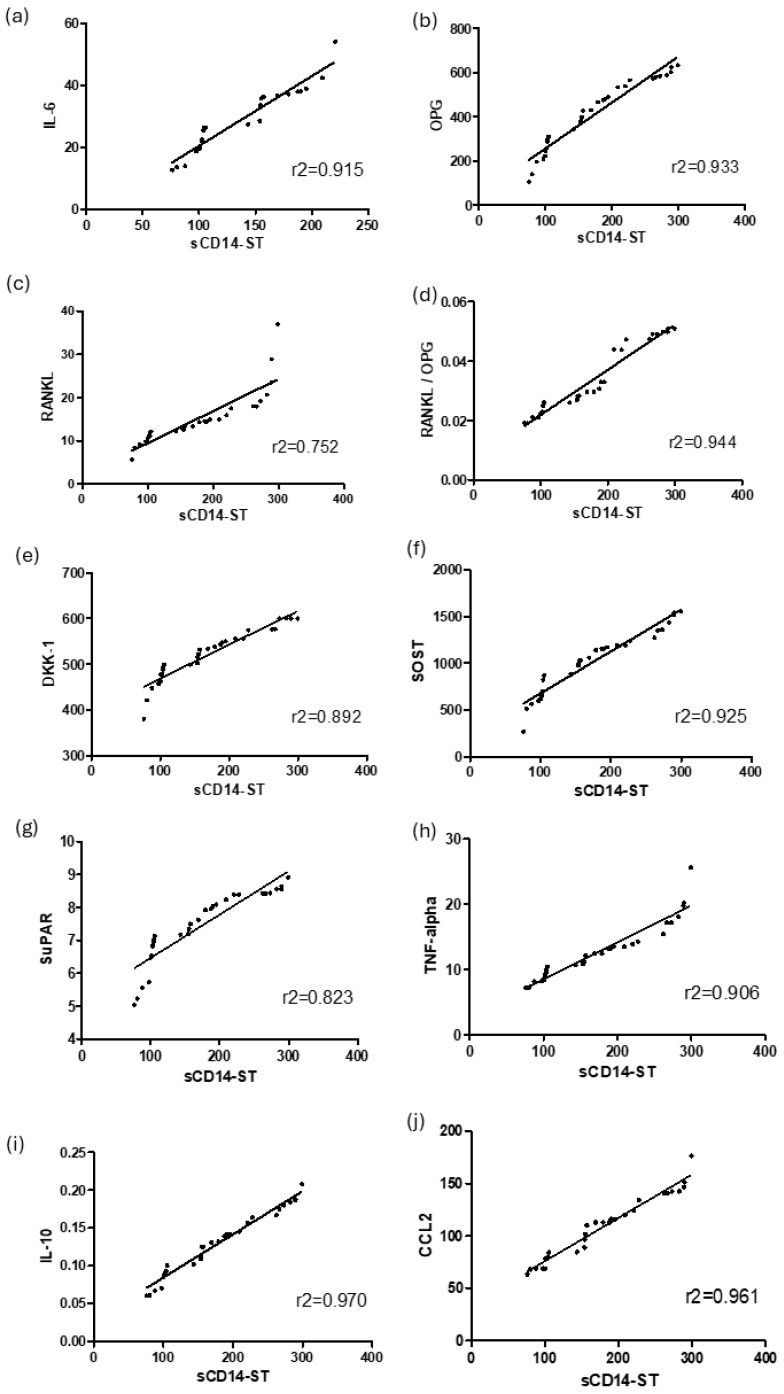
sCD14-ST correlation with inflammatory and osteoimmunological biomarkers. Correlation (Spearman’s r; 95% confidence interval) of sCD14-ST with the other inflammatory and osteoimmunological biomarkers: IL -6 (**a**), OPG (**b**), RANKL (**c**), RANKL/OPG ratio (**d**), DKK-1 (**e**), SOST (**f**), SuPAR (**g**), TNF-alpha (**h**), IL-10 (**i**), and CCL2 (**j**).

**Figure 5 diagnostics-14-01588-f005:**
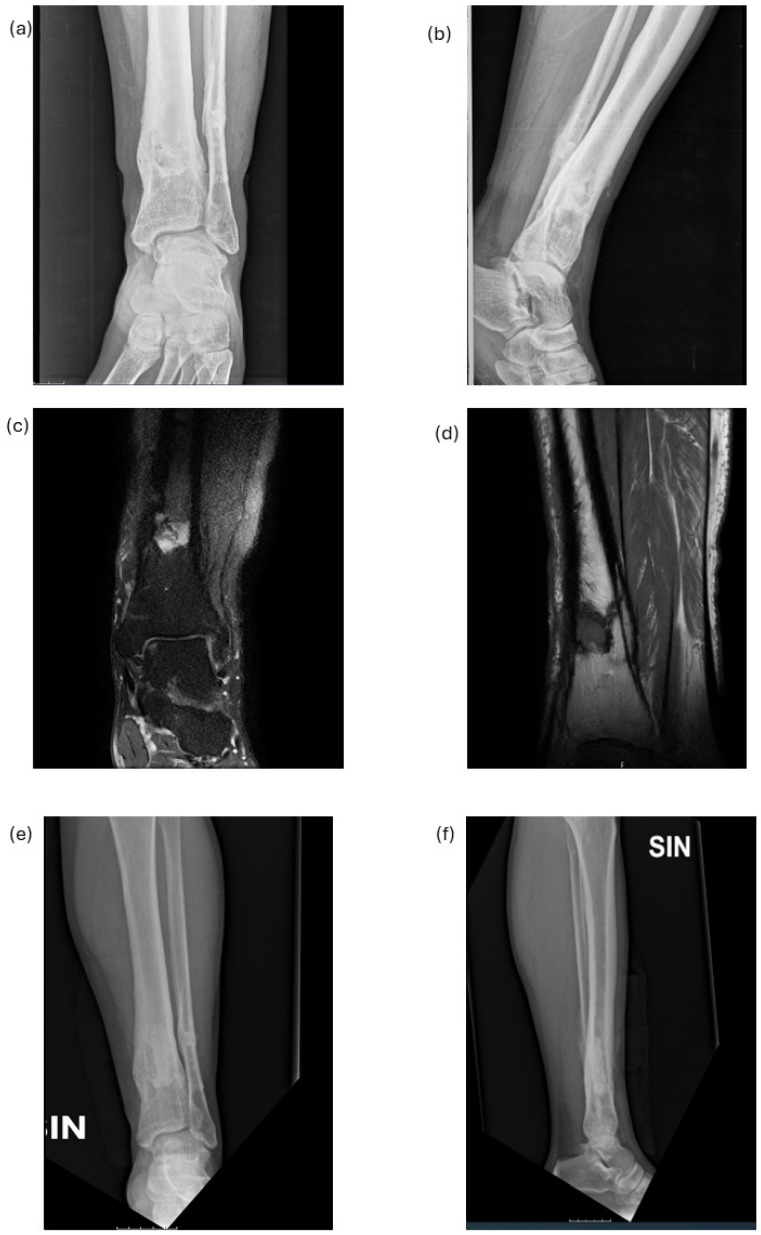
Pre-operative and post-operative radiographic and MRI images of chronic osteomyelitis of distal tibia. Pre-operative radiographic AP (anteroposterior view, (**a**)) and LL (lateral view, (**b**)) images, and MRI AP (**c**) and MRI LL (**d**) images; post-operative radiographic AP (**e**) and LL (**f**) images.

## Data Availability

Data are unavailable due to privacy restrictions.

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
