# Peer review of "sCD14-ST and Related Osteoimmunological Biomarkers: A New Diagnostic Approach to Osteomyelitis"

_diagnostics, 2024, doi:10.3390/diagnostics14151588_

Round 1
Reviewer 1 Report
Comments and Suggestions for Authors
In this clinical study the Authors investigated the levels and role of potential diagnostic biomarkers, especially sCD14ST, in osteomyelitis.
1) In the introductory text, I suggest that the Authors better organize the paragraphs and more clearly address the key points of the study such as the pathophysiology of osteomyelitis (OM), immune system, diagnostic tools and biomarkers, as well as provide an overview of the current literature on the role of potential biomarkers investigated;
2) Regarding the methodology, clarify and further detail the study design in relation to the eligibility criteria, how the Authors considered other factors and associated diseases/conditions that may affect the immune system and bone metabolism. Additionally, add more detailed information about procedures with participants;
3) Considering data available in the literature, I suggest that the Authors better discuss microbiological aspects, immune response and impact on bone tissue, OM stages and marker levels, as well as differences related to individual characteristics (such as sex and age) and status of patient's general health (e.g. metabolic diseases such as obesity and diabetes) and characterization of OM (type of infection and affected site, time and stage of OM);
4) Other considerations: The clarity and structuring of the text in general can be improved by avoiding very short sentences/paragraphs interspersed with very long sentences/paragraphs, for example.
Comments on the Quality of English Language
Minor corrections are necessary to refine the text.
Author Response
Journal Diagnostics (ISSN 2075-4418)
Manuscript ID : diagnostics-3027332
Type :Article
Title : sCD14-ST and related osteoimmunological biomarkers: a new diagnostic approach to osteomyelitis
Reply to Reviewer 1
In this clinical study the Authors investigated the levels and role of potential diagnostic biomarkers, especially sCD14ST, in osteomyelitis.
1) In the introductory text, I suggest that the Authors better organize the paragraphs and more clearly address the key points of the study such as the pathophysiology of osteomyelitis (OM), immune system, diagnostic tools and biomarkers, as well as provide an overview of the current literature on the role of potential biomarkers investigated;
Reply : the Introduction was extensively modified addressing the reviewer’s request , as reported in the text
“ The immune response to OM involves a complex interplay between pro-inflammatory and anti-inflammatory signals. The infection triggers an inflammatory response characterized by the release of cytokines such as IL-6 and TNF-alpha, which recruits immune cells to the infection site. OM is defined as an inflammatory condition of the bone, commonly caused by infection. The infection triggers an inflammatory response that disrupts bone remodeling, resulting in excessive bone resorption and subsequent bone loss. The interaction between the skeletal and immune systems, known as osteoimmunology, plays a crucial role in OM pathogenesis. Inflammatory conditions have been shown to directly affect osteoclast function, leading to imbalanced bone turnover and bone loss. The chronicity of the infection and sustained inflammation play a pivotal role in this panorama. OM progresses through various stages, each characterized by different clinical presentations and biomarker profiles. Acute infection stage marked by severe pain, fever, and elevated levels of inflammatory markers like CRP and IL-6. Subacute stage is characterized by less pronounced systemic symptoms but persistent localized inflammation. Biomarkers such as sCD14-ST and CCL2 may show significant elevation. Long-standing infections have periods of quiescence and exacerbation. Chronic inflammation leads to the formation of sequestra and extensive bone damage. Biomarkers like SuPAR and the RANKL/OPG ratio are often elevated in this phase. Individual features have also an impact on OM development. Indeed, the incidence and severity of OM may differ between males and females due to hormonal influences on the immune response and bone metabolism. Young and elderly patients are more susceptible to OM due to differences in immune system function and bone remodeling rates. In the latest, conditions like diabetes and obesity significantly impact the progression and management of OM. Diabetic patients often have impaired immune responses and poor vascularization, leading to chronic infections and delayed healing. Obesity is associated with systemic inflammation, which can exacerbate the severity of OM. Moreover, immunosuppressive conditions and treatments can predispose individuals to OM and complicate treatment efficacy. Thus, early diagnosis and treatment are crucial for preventing progression from acute to chronic OM. Biomarker levels vary across different stages, providing valuable diagnostic and prognostic information [6,7]. Emerging biomarkers such as soluble CD14 subtype (sCD14ST) have shown promise in diagnosing infections. sCD14-ST is a soluble form of a glycoprotein expressed on monocytes and macrophages, released into circulation in response to pro-inflammatory signals. This study aims to investigate the diagnostic potential of a panel of osteoimmunological and inflammatory serum biomarkers, focusing on sCD14-ST. Recent studies highlight the diagnostic and prognostic potential of sCD14-ST in various infections, including pulmonary infections, COVID-19, and sepsis [10]. Its role in orthopedic infections, such as prosthetic joint infection [11], has been explored, but its application in OM remains under-investigated. “
2) Regarding the methodology, clarify and further detail the study design in relation to the eligibility criteria, how the Authors considered other factors and associated diseases/conditions that may affect the immune system and bone metabolism. Additionally, add more detailed information about procedures with participants;
Reply : According to the reviewer’s suggestion, further details about other factors and associated diseases/conditions that may affect the immune system and bone metabolism were added in the introduction section: “Indeed, the incidence and severity of OM may differ between males and females due to hormonal influences on the immune response and bone metabolism. Young and elderly patients are more susceptible to OM due to differences in immune system function and bone remodeling rates. In the latest, conditions like diabetes and obesity significantly im-pact the progression and management of OM. Diabetic patients often have impaired immune responses and poor vascularization, leading to chronic infections and delayed healing. Obesity is associated with systemic inflammation, which can exacerbate the severity of OM. Moreover, immunosuppressive conditions and treatments can predispose individuals to OM and complicate treatment efficacy.”
In the Methods session, eligibility criteria and procedures with participants were added as follows:
“Inclusion criteria were : age greater than or equal to 18 years, diagnosis of osteomyelitis, subjects of both sexes, and signature of informed consent. Exclusion criteria were the presence of autoimmune diseases, tumors, immunosuppression, or steroid therapy that could be confounding factors by affecting the inflammatory status of the patients even in the absence of a bacterial infection.
Diagnosis of osteomyelitis will be performed via:
- Routine preoperative radiological investigations (X-ray and MRI)
- Laboratory tests required by clinical routine:
Pre-operative: -CRP, complete blood count with leukocyte formula
First post-operative blood sample (1 post-operative day): CRP, complete blood count with leukocyte count
Second post-operative blood sample, if performed (3 weeks post-operative +/- 1 day): CRP, complete blood count with leukocyte count
The determinations were collected by recording the values of the blood samples taken as part of the checks conducted in the normal diagnostic flow. The study did not involve genetic or diagnostic investigations, and any residual samples at the end of the analysis were be destroyed at the end of the study.
Safety and evaluation of adverse reactions in relation to the techniques used: Participating subjects will be closely monitored for any undesirable effects. Any possible reaction was recorded in the appropriate section of the CRF.
Data collection for withdrawn subjects: Even if subjects could withdraw from the study, side effects would have been recorded and the data collected up to that point retained. No side effects were registered. Samples not analyzed or residual was eliminated.
Risk/Benefit Ratio: No specific risks associated with participation in this study have been identified. Clinical and imaging data were collected only from routinely performed evaluations.
The patient did not have direct benefits from participation in the study, but it allows the collection of data useful for the knowledge and diagnosis of the pathology under consideration.
SURGICAL PROCEDURES
“The control group ( OM negative ) was composed of orthopedic patients with no history of osteoarticular infection who were candidates for elective surgery for first implant shoulder, hip, or knee prostheses or arthroscopic shoulder or knee surgery.
For OM-positive patients, elective surgical treatment of sequestrectomy and debridement of the osteomyelitis focus. The surgical procedure consisted of a skin incision targeted to the pathologic portion of bone, followed by skeletonization of the bone, the opening of bone window with an appropriate saw to access the medullary canal, and subsequent thorough cleaning of the canal with the removal of any bone sequestration. All patients received antibiotic therapy for 4-6 weeks (at least 2 of which were intravenous) starting from the day of surgery. Drugs were chosen in consultation with the infectious disease specialist, based on prior isolations (if available) and subsequently modified if necessary, in light of intraoperative culture results. Healing was assessed, upon discontinuation of antibiotic therapy, based on clinical course and normalization of inflammatory indices. No patients experienced recurrence at the current minimum follow-up of 6 months (maximum 2 years). “
3) Considering data available in the literature, I suggest that the Authors better discuss microbiological aspects, immune response and impact on bone tissue, OM stages and marker levels, as well as differences related to individual characteristics (such as sex and age) and status of patient's general health (e.g. metabolic diseases such as obesity and diabetes) and characterization of OM (type of infection and affected site, time and stage of OM);
Reply : the different aspects underlined by the reviewer were addressed improving both the Introduction and Discussion sections, describing OM stages and marker levels, as well as differences related to individual characteristics (such as sex and age) and status of patient's general health (e.g. metabolic diseases such as obesity and diabetes) and characterization of OM (type of infection and affected site, time and stage of OM), as follows:
“OM progresses through various stages, each characterized by different clinical presentations and biomarker profiles. Acute infection stage marked by severe pain, fever, and elevated levels of inflammatory markers like CRP and IL-6. Subacute stage is characterized by less pronounced systemic symptoms but persistent localized inflammation. Biomarkers such as sCD14-ST and CCL2 may show significant elevation. Long-standing infections have periods of quiescence and exacerbation. Chronic inflammation leads to the formation of sequestra and extensive bone damage. Biomarkers like SuPAR and the RANKL/OPG ratio are often elevated in this phase. Individual features have also an impact on OM development. Indeed, the incidence and severity of OM may differ between males and females due to hormonal influences on the immune response and bone metabolism. Young and elderly patients are more susceptible to OM due to differences in immune system function and bone remodeling rates. In the latest, conditions like diabetes and obesity significantly impact the progression and management of OM. Diabetic patients often have impaired immune responses and poor vascularization, leading to chronic infections and delayed healing. Obesity is associated with systemic inflammation, which can exacerbate the severity of OM. Moreover, immunosuppressive conditions and treatments can predispose individuals to OM and complicate treatment efficacy. Thus, early diagnosis and treatment are crucial for preventing progression from acute to chronic OM. Biomarker levels vary across different stages, providing valuable diagnostic and prognostic information [6,7]. Emerging biomarkers such as soluble CD14 subtype (sCD14ST) have shown promise in diagnosing infections. sCD14-ST is a soluble form of a glycoprotein expressed on monocytes and macrophages, released into circulation in response to pro-inflammatory signals.”
Discussion section was also improved by adding information about microbiological aspects, immune response and impact on bone tissue, OM stages and marker levels, as follows:
“Osteomyelitis, or inflammation of bone, is most commonly caused by invasion of bacterial pathogens into the skeleton. Bacterial osteomyelitis treatment is difficult to treat, due to widespread antimicrobial resistance in the preeminent etiologic agent, the Gram-positive bacterium Staphylococcus aureus. Bacterial osteomyelitis triggers pathological bone remodeling, which in turn leads to sequestration of infectious foci from innate immune effectors and systemically delivered antimicrobials. Treatment of osteomyelitis therefore typically consists of long courses of antibiotics associated with surgical debridement of necrotic infected tissues.
A standard approach for diagnosing osteomyelitis is based on clinical suspicion. The initial evaluation includes measuring serum C-reactive protein (CRP) and erythrocyte sedimentation rate (ESR), radiographs, and blood cultures (19) Elevated ESR and CRP levels are nonspecific indicators for osteomyelitis, as different inflammatory conditions can enhance ESR and CRP levels. In cases of acute osteomyelitis, patients often have normal ESR and CRP levels, which may decrease in chronic OM. Our patients had chronic OM, so is not surprising that traditional inflammatory markers like CRP and PCT were not significantly elevated compared to non-infected patients. It is described that bone turnover decreases with age, shifting the balance towards bone loss. This process also affects osteoimmunological factors. Inclusion criteria excluded pediatric patients, who can present specific characteristics of OM, in order to focus only on adult ones.
A recent study [Nair S, Hatkar S, Patil A, et al. Age-related changes and reference intervals of RANKL, OPG, and bone turnover markers in Indian women. Arch Osteoporos. 2021;16(1):146. doi: 10.1007/s11657-021-01014-4. PMID: 34606009 ] reported reference intervals for serum RANKL, OPG, RANKL/OPG ratio based on menopausal status and age. It found that median RANKL RANKL/OPG were higher, while OPG levels were lower in premenopausal women compared to postmenopausal women. Also host response is known to decrease with age according to changes in immunological function, termed «immunosenescence». This includes a diminished immune response against infections, increased levels of pro-inflammatory mediators, and a higher risk of autoimmunity. Immunoscenescence is accompanied by low-grade aseptic inflammation, commonly observed in elderly people, defined «inflammaging». This process affects osteoimmunological factors by increasing RANKL levels relative to OPG levels, thus shifting the balance towards bone loss. To exclude age or sex confounding factors, both study groups (OM positive and negative) were age and sex matched. “
.
4) Other considerations: The clarity and structuring of the text in general can be improved by avoiding very short sentences/paragraphs interspersed with very long sentences/paragraphs, for example
Reply : The text structure was corrected following the reviewer ‘s suggestions

Reviewer 2 Report
Comments and Suggestions for Authors
Dear Editor and Authors,
the authors present an interesting study investigating the potential diagnostic role of osteoimmunological serum biomarkers in the Osteomyelitis (OM) clinical approach. The results indicate that sCD14-ST has a diagnostic an early prognostic role in OM patients. A diagnostic role could be observed too for SuPAR, CCL2, IL-10, the Wnt inhibitors DKK-1 and Sclerostin and RANKL / OPG ratio, whereas CCL2 and SuPAR also show a prognostic role. The emerging role of sCD14-ST as a new tool for OM diagnosis and prognosis was also confirmed in correlation with other biomarkers.
Overall, the paper is well written. The opportunity to diagnosis OM patients early
is a promising approach. There are some minor concerns before considering the publication of this study.
Introduction:
§ How specific is sCD14-ST for OM if it is also elevated in other infections such as SARS-CoV-2, which can also be clinically inapparent?
Patients & Methods
§ The study mainly included "older" patients. Is there any evidence that the osteoimmunoligical profile changes with age? Or whether the osteoimmunoligic host reacts differently to infections in older patients?
§ OM is often a problem in immune compromised patients, but these patients were excluded. Is this study design not a significant limitation for the evaluation of sCD14-ST?
§ Line 227, page 5: Spelling error "negative e patients"
§ Did the OM patients included in the study have metal implants? Was it investigated whether metal implants have an influence on the biomarkers?
Discussion:
§ Are there any data on increasing sCD14-ST during biofilm formation? Are there differences in the biomarkers examined in patients who were detected to have biofilm formation?
§ The regulation of osteoimmunological genes is complex and only a few pathways have been investigated. Would it not be necessary to correlate the systemically investigated biomarkers with local samples of OM (investigations at the gene expression or protein level) to be able to reliably prove correlations?
§ The biomarkers described are also modified in non-unions. Does research exist on these biomarkers to differentiate between aseptic and septic non-unions?
§ Line 385, page 12: spelling mistake "marker instead of maker"
With kind regards
Author Response
Introduction:
- How specific is sCD14-ST for OM if it is also elevated in other infections such as SARS-CoV-2, which can also be clinically inapparent?
REPLY : sCD14ST is not specific only for OM but it can be considered a biomarker of phagocytosis of whole bacterial cells, including both Gram-negative and Gram-positive bacteria, and it is successfully used in clinical practice as a biomarker for bacterial infections and sepsis [Memar MY, Baghi HB. Presepsin: A promising biomarker for the detection of bacterial infections. Biomed Pharmacother. 2019;111:649-656. doi: 10.1016/j.biopha.2018.12.124. PMID: 30611989.; Yang HS, Hur M, Yi A, et al. Prognostic value of presepsin in adult patients with sepsis: Systematic review and meta-analysis. PLoS One. 2018;13(1):e0191486. doi: 10.1371/journal.pone.0191486. PMID: 29364941; de Guadiana Romualdo LG, Torrella PE, Acebes SR, et al. Diagnostic accuracy of presepsin (sCD14-ST) as a biomarker of infection and sepsis in the emergency department. Clin Chim Acta. 2017;464:6-11. doi: 10.1016/j.cca.2016.11.003. PMID: 27823951.; Chenevier-Gobeaux C, Borderie D, Weiss N, et al. Presepsin (sCD14-ST), an innate immune response marker in sepsis. Clin Chim Acta. 2015;450:97-103. doi: 10.1016/j.cca.2015.06.026. PMID: 26164388; Papp M, Tornai T, Vitalis Z, et al. Presepsin teardown - pitfalls of biomarkers in the diagnosis and prognosis of bacterial infection in cirrhosis. World J Gastroenterol. 2016;22(41):9172-9185. doi: 10.3748/wjg.v22.i41.9172. PMID: 27895404].
Recent evidence also indicates that, even in the absence of obvious infection, presepsin can also be used as a universal biomarker of cellular bacterial translocation in cirrhosis [Efremova I, Maslennikov R, Poluektova E, et al. Presepsin as a biomarker of bacterial translocation and an indicator for the prescription of probiotics in cirrhosis. World J Hepatol. 2024;16(5):822-831. doi: 10.4254/wjh.v16.i5.822. PMID: 38818295].
sCD14ST levels can be considered an indicator of activation in the innate immune response to pathogen invasion. Plasma levels of sCD14ST can be considered an indicator of activating innate immune cells in response to invading pathogens [Memar MY, Baghi HB. Presepsin: A promising biomarker for the detection of bacterial infections. Biomed Pharmacother. 2019;111:649-656. doi: 10.1016/j.biopha.2018.12.124. PMID: 30611989.; Chenevier-Gobeaux C, Borderie D, Weiss N, et al. Presepsin (sCD14-ST), an innate immune response marker in sepsis. Clin Chim Acta. 2015;450:97-103. doi: 10.1016/j.cca.2015.06.026. PMID: 26164388.; Liu B, Chen YX, Yin Q, et al. Diagnostic value and prognostic evaluation of Presepsin for sepsis in an emergency department. Crit Care. 2013;17(5):R244. doi: 10.1186/cc13070. PMID: 24138799].
The sCD14ST levels are significantly elevated in severe COVID-19 patients compared to mild cases, as confirmed by a recent meta-analysis study that indicated sCD14ST as a promising biomarker that can accurately reflect the severity of COVID-19 [Guarino M, Perna B, Maritati M, et al. Presepsin levels and COVID-19 severity: a systematic review and meta-analysis. Clin Exp Med. 2023;23(4):993-1002. doi: 10.1007/s10238-022-00936-8. PMID: 36380007;]. Since sCD14ST can reflect the severity of patients’ inflammatory response, it may also have a prognostic role in COVID-19 outcome, as recently described by our group [Galliera E, Massaccesi L, Yu L, et al. SCD14-ST and New Generation Inflammatory Biomarkers in the Prediction of COVID-19 Outcome. Biomolecules. 2022;12(6):826. doi: 10.3390/biom12060826. PMID: 35740951]
Based on these premises, while sCD14-ST can be a useful biomarker for detecting bacterial infections, its lack of specificity due to elevation in various infectious and inflammatory conditions, including SARS-COV-2, necessitates a comprehensive diagnostic approach. Relying solely on sCD14-ST levels for diagnosing OM would be inappropriate and could lead to misdiagnosis or oversight of other conditions. However, it should be part of a broader diagnostic approach that includes clinical evaluation, diagnostic imaging, microbiological testing, and other inflammatory markers such as C-reactive protein (CRP) and procalcitonin (PCT).
Patients & Methods
- The study mainly included "older" patients. Is there any evidence that the osteoimmunoligical profile changes with age? Or whether the osteoimmunoligic host reacts differently to infections in older patients?
REPLY : It is described that bone turnover decreases with age, shifting the balance towards bone loss. This process also affects osteoimmunological factors. A recent study [Nair S, Hatkar S, Patil A, et al. Age-related changes and reference intervals of RANKL, OPG, and bone turnover markers in Indian women. Arch Osteoporos. 2021;16(1):146. doi: 10.1007/s11657-021-01014-4. PMID: 34606009] reported reference intervals for serum RANKL, OPG, RANKL/OPG ratio based on menopausal status and age. It found that median RANKL RANKL/OPG were higher, while OPG levels were lower in premenopausal women compared to postmenopausal women. Also host response is known to decrease with age according to changes in immunological function, termed «immunosenescence». This includes a diminished immune response against infection, increased levels of pro-inflammatory mediators, and a higher risk of autoimmunity. Immunoscenescence is accompanied by low-grade aseptic inflammation, commonly observed in elderly people, defined «inflammaging». This process affects osteoimmunological factors by increasing RANKL levels relative to OPG levels, thus shifting the balance towards bone loss.
To exclude age or sex confounding factors, both study groups (OM positive and negative) were age and sex matched.
- OM is often a problem in immune compromised patients, but these patients were excluded. Is this study design not a significant limitation for the evaluation of sCD14-ST?
REPLY : To our knowledge, this is the first study evaluating sCD14ST in OM. To clarify the potential diagnostic role of this marker in OM, we exclude additional alterations of immune system, such as autoimmunity, which could act as a confounding factor in this initial study. The author really appreciate this suggestion, and once the diagnostic and prognostic roles of sCD14ST in OM are defined, further studies will extend the use of sCD14ST in OM patients with autoimmune disorders.
- Line 227, page 5: Spelling error "negative epatients"
REPLY: the spelling error was corrected as indicated
- Did the OM patients included in the study have metal implants? Was it investigated whether metal implants have an influence on the biomarkers?
Reply : Reply : OM patients included in the study did not have metal implants, thus the influence of metal implants on the biomarkers was not investigated in this study. However, our group recently showed that sCD14ST is a useful tool for the diagnosis and clinical monitoring of Prosthetic Joint Infections. This can be supported by a panel of new inflammatory markers involved in the monocyte/macrophage-mediated inflammatory response [Monica Gioia Marazzi , Filippo Randelli , Marco Brioschi et Al 1 3Int J Immunopathol Pharmacol 2018 Jan-Dec:31:394632017749356. doi: 10.1177/0394632017749356. Epub 2017 Dec 18.Presepsin: A potential biomarker of PJI? A comparative analysis with known and new infection biomarkers].
Discussion:
- Are there any data on increasing sCD14-ST during biofilm formation? Are there differences in the biomarkers examined in patients who were detected to have biofilm formation?
REPLY : Biofilm is present in sequestrative osteomyelitis as the dead bone or sequestra behaves like an inert body and therefore induces the formation of biofilm.
All the OM patients in the study display chronic OM, characterized by necrotic bone in the sequestrum area. The surgical approach was directed to the removal of necrotic bone. Since all the OM patients of the study display chronic OM, mainly due to Staphylococcus Aureus infection, the main pathogenic bacteria of chronic osteomyelitis forming biofilm, it is reasonable to hypothesize the presence of a biofilm in our OM patients.
For this reason, in addition to the surgical debridement, pre and post-surgical antibiotic treatment was performed, to target any possible bacterial biofilm.
In periodontitis, oral biofilm challenge has been reported to regulate the RANKL-OPG system in periodontal ligament and dental pulp cells, but no data are currently available on sCD14ST during biofilm formation.
In this study it was not possible to evaluate the possible increase in sCD14ST or the other biomarkers investigated as these are chronic stabilized infections and therefore the biofilm was already formed and stabilized. .
- The regulation of osteoimmunological genes is complex and only a few pathways have been investigated. Would it not be necessary to correlate the systemically investigated biomarkers with local samples of OM (investigations at the gene expression or protein level) to be able to reliably prove correlations?
Reply : The aim of the study was to explore the potential diagnostic role of the serum sCD14ST levels in OM diagnosis. The choice of a serum maker is directed to find a non-invasive approach for patients that are monitored several time during the pre-post surgical period. The diagnostic potential of sCD14ST was based on the correlation between its circulating levels and the clinical tools currently used for OM diagnosis. Understanding the regulatory mechanisms is certainly more complex and the evaluation of gene expression on protein levels in local OM samples will be addressed in future studies to provide new insights into the the regulation of osteoimmunolgical pathways. The authors appreciate the suggestion and will incorporate it into our future research plans.
- The biomarkers described are also modified in non-unions. Does research exist on these biomarkers to differentiate between aseptic and septic non-unions?
Reply : RANKL/OPG ratio has been described having a role in osteoclast-mediated nonunion disorders. To the best of the authors’ knowledge, there is currently no evidence showing that the described biomarkers can differentiate between septic and aseptic nonunions. Further investigation in this field is needed to provide new insights.
- Line 385, page 12: spelling mistake "marker instead of maker"
REPLY: the spelling mistake was corrected as indicated

Reviewer 3 Report
Comments and Suggestions for Authors
sCD14ST has been subject to several studies about its usefulness in diagnosing different infections including COVID-19 and post-surgical spinal infection. The authors investigated the role of sCD14ST in the diagnosis of osteomyelitis. They showed that sCD14ST can be a useful marker for OM.
The paper can benefit from some minor revisions
1. Did all patients with OM have acute OM? Is there any difference regarding sCD14ST levels between patients with acute and chronic OM?
2. How did the authors decide on the required number of patients for this study?
3. “OM confirmed by radiographic test , at… “ Please, clear the type of radiographic test.
4. The underlined bacteria species should be corrected.
‘ …while only 3 patients displayed an 89 coinfection of Staphylococcus aureus and Staphylococcus agalactia and only one patient 90 showed a coinfection of Staphylococcus agalactia and Enterobacter cloache”
Comments on the Quality of English Language
Minor editing can be useful
Author Response
Journal Diagnostics (ISSN 2075-4418)
Manuscript ID : diagnostics-3027332
Type :Article
Title : sCD14-ST and related osteoimmunological biomarkers: a new diagnostic approach to osteomyelitis
Reply to reviewer 3
The paper can benefit from some minor revisions
- Did all patients with OM have acute OM?
Reply : According to Cierny & Mader anatomo-pathological classification, 18 patients were classified as Type 1 osteomyelitis, 2 as Type 2, 22 as Type 3. Host type based on comorbidities was classified according to Cierny & Made classification prevalent as Type A and Type B.
All patients underwent pre-operative clinical and laboratory tests evaluation, x-ray, CT and MRI scan. Concerning clinical presentation and local inflammatory signs (redness, swelling, pain, local warmth), all patients were considered to have a chronic infection (presence of only one or no signs of local inflammation). 17 patients had a draining sinus at the time of surgery.
- Is there any difference regarding sCD14ST levels between patients with acute and chronic OM?
Reply : no significative difference in sCD14ST was detected among OM patients
- How did the authors decide on the required number of patients for this study?
Reply : The sample size was calculated using a dedicated statistical program available online (http://www.sample-size.net/sample-size-means/), considering a Type I and II error rate equal to 0.05 and 0, respectively, with an effect size of 0.7 and a standard deviation of 1. This calculation yielded 34 patients per group. Considering a dropout of approximately 20%, the estimated number of patients was 40 per group, resulting in a total of 80 patients enrolled. Due to the expected dropout rate, patient enrollment was extended until the calculated sample size was reached. The study included 42 patients in the OM group and 35 patients in the OM negative group (with five OM negative patients initially enrolled being excluded during the study for not meeting the inclusion criteria).
- “OM confirmed by radiographic test , at… “ Please, clear the type of radiographic test.
Reply : The sentence was replaced with a a more detailed description of diagnostic approach : “All patients underwent pre-operative clinical and laboratory tests evaluation, x-ray, CT and MRI scan”
- The underlined bacteria species should be corrected.
‘ …while only 3 patients displayed an 89 coinfection of Staphylococcus aureus and Staphylococcus agalactia and only one patient 90 showed a coinfection of Staphylococcus agalactia and Enterobacter cloache”
Reply : the bacteria species were corrected as suggested
…while only 3 patients displayed a coinfection of Staphylococcus aureus and Staphylococcus agalactiae and only one patient showed a coinfection of Staphylococcus agalactiae and Enterobacter cloacae”

Round 2
Reviewer 1 Report
Comments and Suggestions for Authors
The adjustments complemented and improved the manuscript.
Author Response
Dear Authors,
I accept the current format, and kindly ask you to add a minor revision of the Limitations' sub-section.At Limitations sub-section, please pinpoint the followings (in addition to the short time frame of the study that has already been mentioned):
-larger, multi-centers studies might help the conclusions of the current study for practical purposes
-a potential source of bias might come from a large panel of co-morbidities and prior medications that might act as confounding factors amid the present analysis
-further studies will show the pathogenic traits underlying the current clinical data
Reply :
The authors appreciated the referee’s suggestion, to better explain the limitations of the present study and the future perspective to overcome them.
The text was modified by adding the aspects suggested by the reviewer, as follows :
“ The limitation of this study is the short period of longitudinal evaluation, conducted only at a close time point after surgery according to clinical protocols during patients’ hospitalization. In addition , a potential source of bias might come from a large panel of comorbidities and prior medications that might act as confounding factors amid the present analysis. Biomarkers that displayed a faster response to infection eradication showed a decrease in OM patients even at this short post-surgical time point. However, a longer longitudinal evaluation with additional time points could provide more information about the prognostic potential of biomarkers that remained stable shortly after surgery. Further studies will show the pathogenic traits underlying the current clinical data and larger, multi-center studies might help the conclusions of the current study for practical purposes”